# CD133 Expression in Circulating Tumor Cells as a Prognostic Marker in Colorectal Cancer

**DOI:** 10.3390/ijms26104740

**Published:** 2025-05-15

**Authors:** Katsuji Sawai, Kenji Koneri, Youhei Kimura, Takanori Goi

**Affiliations:** First Department of Surgery, University of Fukui, Fukui 910-1193, Japan; y-kimura@kimura-hospital.jp (Y.K.); tgoi@u-fukui.ac.jp (T.G.)

**Keywords:** colorectal adenocarcinoma, CD133, cancer stem cells, circulating tumor cells, tumor marker

## Abstract

Identifying prognostic markers in colorectal cancer (CRC) is crucial for improving treatment outcomes. Although carcinoembryonic antigen (CEA) is recommended in the guidelines of the National Comprehensive Cancer Network, its sensitivity and specificity are inconsistent, limiting its utility in patients with normal CEA levels. Circulating tumor cells (CTCs), including those expressing CD133—a cancer stem cell marker involved in tumor progression and therapy resistance—are associated with metastasis and survival outcomes. This study evaluated the prognostic significance of CD133-positive CTCs, and their combined effect with CEA, in patients with CRC. Peripheral blood samples from 195 patients with CRC (stages I–IV) were analyzed. CTCs were isolated using OncoQuick tubes and CD133 mRNA expression was detected by reverse transcription polymerase chain reaction. In clinicopathological analysis, CD133-positive CTCs were detected in 27.2% of cases, correlating with serosal invasion (*p* = 0.016). Multivariate Cox analysis showed that CD133-positive CTCs were associated with worse disease-specific survival (*p* = 0.001). Patients with CD133-positive CTCs and CEA ≥ 5 ng/mL (high CEA) had a significantly poorer prognosis (*p* < 0.001), whereas those with CD133-negative CTCs and CEA < 5 ng/mL (low CEA) had a better prognosis (*p* = 0.039). CD133 expression in CTCs, especially in combination with CEA, may serve as a valuable prognostic marker in CRC.

## 1. Introduction

Colorectal cancer (CRC) is the third most diagnosed cancer worldwide, with approximately 1.9 million new cases and 903,859 deaths annually [1]. CRC accounts for a significant proportion of cancer-related mortality, with recurrence and metastasis being the primary contributors to poor outcomes [2]. Although recent advancements in molecular targeted therapies and immunotherapies have improved survival rates, these treatments are often associated with substantial financial costs, adverse effects, and limited availability [3]. These challenges underscore the urgent need for reliable biomarkers that can predict recurrence, assess prognosis, and optimize treatment strategies.

Carcinoembryonic antigen (CEA) is one of the most widely used tumor markers in CRC, and is recommended in the guidelines of the National Comprehensive Cancer Network (NCCN) for prognostic assessment and disease monitoring [4,5,6,7,8,9]. Elevated preoperative CEA levels have been consistently associated with poor prognosis and an increased risk of recurrence. Despite its utility, the clinical application of CEA levels is limited by inconsistent cutoff values, sensitivity, and specificity, which result in false positives and negatives in certain patient populations [3,10,11,12]. This variability highlights the need for novel biomarkers that can complement or surpass the prognostic accuracy of CEA.

Metastasis, the leading cause of CRC-related mortality, is a complex process involving multiple steps, including tumor cell detachment, intravasation into the bloodstream, survival as circulating tumor cells (CTCs), and colonization of distant organs [13]. Elevated CTC counts have been associated with worse progression-free survival (PFS) and overall survival (OS) in patients with CRC, suggesting their potential as independent prognostic markers [14,15,16]. Furthermore, dynamic changes in CTC counts during treatment have been proposed as early indicators of therapeutic efficacy, particularly in metastatic CRC [17]. However, the clinical utility of CTC enumeration is limited by the phenomenon of “metastatic inefficiency”, wherein only a small fraction of CTCs possess the capacity to form metastatic lesions [18]. Understanding the biological properties of these rare, metastasis-competent CTCs is critical for improving prognostic and therapeutic strategies.

Emerging evidence suggests that a subpopulation of CTCs, with cancer stem cell (CSC)-like properties, play a pivotal role in metastasis, recurrence, and resistance to therapy [19,20,21]. CSCs are a small subset of tumor cells, and possess stem cell-like features that include self-renewal, differentiation, and enhanced resistance to conventional therapies. In CRC, several CSC-specific markers, such as leucine-rich repeat-containing G-protein-coupled receptor 5 (LGR5), CD44, and CD133, have been identified as indicators of tumor aggressiveness and poor clinical outcomes [22,23,24]. Among these, CD133 has emerged as one of the most promising markers due to its association with tumorigenicity, chemoresistance, and metastatic potential [25]. CD133, also known as Prominin-1, is a transmembrane glycoprotein predominantly expressed on the apical membrane protrusions of embryonic epithelial cells and various stem-like cancer cells [20,26]. CD133-positive CRC cells exhibit enhanced self-renewal, invasive capacity, and chemoresistance compared to CD133-negative CRC cells [27,28,29]. These properties are mediated through the activation of critical signaling pathways, such as Wnt, Notch, and Hedgehog, which are essential for maintaining CSC characteristics and promoting tumor progression [27,28,29]. Additionally, under hypoxic conditions, CD133 expression has been linked to the upregulation of hypoxia-inducible factor-1α (HIF-1α) and epithelial-mesenchymal transition (EMT) markers, such as N-cadherin and vimentin, facilitating metastatic dissemination [30,31]. Clinical studies have consistently reported that high CD133 expression in primary CRC tumors is associated with advanced disease stages, increased recurrence rates, and poor survival outcomes [32,33,34]. However, the prognostic role of CD133 expression in CTCs is underexplored. While CTC enumeration has demonstrated potential as a prognostic tool, integrating CD133 expression into CTCs with established markers like CEA may provide a more comprehensive understanding of disease progression and patient prognosis. To date, no studies have investigated the synergistic prognostic value of CD133 expression in CTCs and CEA levels in patients with CRC.

In this study, we aimed to evaluate the prognostic significance of CD133 expression in CTCs for disease-specific survival (DSS) in patients with CRC. Secondary objectives included assessing the combined utility of CD133 expression in CTCs and CEA levels for prognostic evaluation and exploring the associations between CD133-positive CTCs and clinicopathological factors, such as tumor stage, grade, and metastatic potential. By addressing these gaps, this study aims to provide novel insights into the biology of CSCs in CRC and highlight the potential of CD133-positive CTCs as a robust biomarker for improving prognostic accuracy and guiding personalized treatment strategies.

## 2. Results

### 2.1. Associations Between CD133 Expression and Clinicopathologic Features

Table 1 presents the baseline demographic and clinicopathological characteristics of the 195 patients with colorectal cancer (CRC) included in this study. The median age at diagnosis was 71 years (range, 38–91 years). According to TNM staging, 44 patients (22.6%) were classified as stage I, 64 (32.8%) as stage II, 54 (27.7%) as stage III, and 33 (16.9%) as stage IV. CD133-positive CTCs were detected in 53 of the 195 cases (27.2%; Figure 1), whereas none of the healthy donors had CD133-positive CTCs. In analysis stratified by stage, CD133-positive CTCs were detected in 5 stage I, 25 stage II, 17 stage III, and 6 stage IV CRC cases. Univariate analysis revealed no correlations between CD133-positive CTCs and tumor size, histological type, lymph node metastasis, distant metastasis, or stage. However, a significant positive correlation was observed between CD133-positive CTCs and serosal involvement. No significant association was observed between serum CEA levels and CD133 expression (Table 1). In multivariate logistic regression analysis, the presence of CD133-positive circulating tumor cells (CTCs) was independently associated with serosal invasion (odds ratio [OR], 3.000; 95% confidence interval [CI], 1.225–7.345; *p* = 0.016). Conversely, no significant associations were found between CD133 expression and other pathological features (Table 2).

### 2.2. Prognostic Impact of CD133 Expression in Circulating Tumor Cells on Disease-Specific Survival

The median observation period for the 195 patients was 44.9 months, during which 31 patients died of the primary disease. The 5-year DSS rate for CD133-positive cases in stages I–III was 85.3%, whereas it was 95.0% for CD133-negative cases (Figure 2). This indicates that CD133-positive cases had a significantly poorer prognosis (*p* = 0.018). However, stratified analyses by stage revealed no statistically significant association between CD133 expression and 5-year DSS in stage I or II cases, nor with 2-year DSS in stage IV cases (*p* = 0.128, 0.721, and 0.275, respectively; Appendix A). Notably, among patients with stage III CRC, those with CD133-positive CTCs had a significantly lower 5-year DSS rate compared to CD133-negative cases (73.8% vs. 96.3%, *p* = 0.021; Appendix A), suggesting a prognostic role of CD133 expression in this subgroup.

### 2.3. Multivariate Cox Analysis for DSS, Incorporating CD133 Expression

Table 3 summarizes the results of the multivariate Cox proportional hazards analysis, which included age, sex, distant metastasis, and CD133 expression as covariates. Both distant metastasis and CD133 expression were found to be independently associated with disease-specific survival (*p* < 0.05 for both).

### 2.4. Prognostic Impact of Preoperative CEA Levels on Disease-Specific Survival

Among the 195 patients, 70 cases exhibited elevated preoperative CEA levels (≥5 ng/mL) distributed across stages I, II, III, and IV as follows: 7, 23, 19, and 21 cases, respectively. In patients with stage I–III colorectal cancer, the 5-year disease-specific survival (DSS) rate was significantly lower in the high CEA group compared to the low CEA group (<5 ng/mL) (76.1% vs. 98.5%, *p* < 0.001; Figure 3). Stage-specific analyses showed no significant association between CEA levels and DSS in stage I (5-year DSS, *p* = 0.270) or stage IV (2-year DSS, *p* = 0.488) cases (Appendix A). In contrast, among stage II patients, the 5-year DSS rate was markedly lower in the high CEA group compared to the low CEA group (78.4% vs. 100.0%, Appendix A). A similar trend was observed in stage III patients (72.8% vs. 100.0%, Appendix A). These findings indicate that elevated preoperative CEA levels are significantly associated with worse prognosis in stage II and III colorectal cancer (*p* = 0.033 and *p* = 0.005, respectively).

### 2.5. Multivariate Cox Analysis for DSS, Incorporating CEA Levels

Table 4 presents the results of the multivariate Cox analysis evaluating age, sex, distant metastasis, and CEA levels for DSS in all patients. The analysis revealed significant associations between DSS and both distant metastasis and elevated CEA levels (both *p* < 0.05).

### 2.6. Impact of Combined CD133 Expression and CEA Levels on DSS in Stage I–III Cases

Among the patients with stage I-III, 83 cases had CD133-negative CTCs with low CEA levels, 32 had CD133-negative CTCs with high CEA levels, 30 had CD133-positive CTCs with low CEA levels, and 17 had CD133-positive CTCs and high CEA levels. The 5-year survival rates for these groups were 100%, 80.0%, 94.4%, and 68.0%, respectively. The differences were significant (*p* < 0.001; Figure 4).

### 2.7. Prognostic Impact of Combined CD133-Positive CTCs and Elevated CEA Levels on Disease-Specific Survival

CD133-positive circulating tumor cells (CTCs) combined with elevated CEA levels (≥5 ng/mL) were identified in 0 of 44 stage I cases, 11 of 64 stage II cases, 6 of 54 stage III cases, and 3 of 33 stage IV cases. When evaluating all colorectal cancer stages collectively, patients with both CD133-positive CTCs and high CEA levels demonstrated significantly poorer 5-year disease-specific survival (DSS) compared to those without this combination (60.4% vs. 80.7%, *p* = 0.014; Figure 5). However, in stage II cases, no statistically significant difference in 5-year DSS was observed between patients with this high-risk profile and other cases (88.9% vs. 94.7%, *p* = 0.127; Appendix A). However, stage III cases with CD133-positive CTCs and high CEA levels showed a significantly worse 5-year DSS (40.0% vs. 96.8%, *p* < 0.001) (Appendix A).

Table 5 presents the results of the multivariate Cox analysis evaluating age, sex, distant metastasis, and CD133-positive CTCs with high CEA levels in all the patients.

Distant metastasis and CD133-positive CTCs with high CEA levels were significantly associated with poorer prognosis (*p* < 0.001 for both).

## 3. Discussion

In this study, we investigated the association between CD133 mRNA expression in CTCs and the prognosis of CRC. Our results revealed that patients with CD133-positive CTCs had a significantly poorer prognosis. Furthermore, cases with CD133-negative CTCs and low CEA levels showed a significantly better prognosis compared to that of other cases. In contrast, patients with CD133-positive CTCs and high CEA levels exhibited significantly poorer prognosis. Additionally, the prevalence of CD133-positive CTCs was significantly higher in cases with serosal invasion, suggesting their potential role in tumor infiltration. CTCs are cancer cells that detach from the primary tumor and enter the bloodstream [35]. Previous studies have reported that higher CTC counts are associated with poorer prognosis [14,15,16]. However, only approximately 1 in 10,000 CTCs is estimated to be capable of initiating metastasis. This phenomenon, termed “metastatic inefficiency”, indicates that while CTCs are necessary for metastasis, they are not sufficient for metastatic progression [18]. Furthermore, CSCs constitute a subset of the CTCs [36,37,38,39]. CSCs possess the capacity for self-renewal and play a crucial role in tumor initiation and progression. They are also implicated in chemoresistance and can differentiate into most tumor cells, which lack tumorigenic potential [40,41,42]. Lee et al. reported that among patients with breast cancer, cases with high expression of CSC markers, including CD133 in CTCs, exhibited poor treatment response and worse OS and PFS [43]. Similarly, Yang et al. demonstrated that among patients with metastatic castration-sensitive prostate cancer, CD133 expression in CTCs was associated with poorer PFS [44]. We have previously reported that among patients with CRC, the expression of CD44 variant exon 9 (CD44v9), a known CSC marker, in CTCs serves as a reliable prognostic marker for poor outcomes [45].

CD133 is considered a prominent CSC marker in CRC, but its functional role remains unclear. It has been reported that CD133-positive cancer cells exhibit a higher proliferative capacity than CD133-negative cancer cells do [27]. Furthermore, CD133 has been implicated in hypoxic adaptation, EMT, and mesenchymal–epithelial transition, which are crucial for cancer metastasis, and in the suppression of apoptosis.

As solid tumors grow, they experience insufficient blood supply, leading to a hypoxic environment. To adapt to this condition, tumors undergo angiogenesis and disseminate through the bloodstream to more favorable environments [46]. A central factor in this process is HIF-1α [31]. HIF-1α not only promotes angiogenesis and cell proliferation but also enhances cancer cell survival and migration, ultimately accelerating tumor invasion and metastasis [47]. Okada et al. reported that in CRC, CD133-positive cells exhibit significantly higher HIF-1α expression under hypoxic conditions than CD133-negative cells do [30]. Consequently, hypoxia is thought to promote EMT in CD133-positive cells. Additionally, in CD133-positive cells, E-cadherin expression, which is downregulated during EMT, has been shown to return to baseline levels within 24 h upon reoxygenation, leading to mesenchymal–epithelial transition and subsequent metastatic formation [30].

Moreover, HIF-1α is known to activate autophagy in CD133-positive cells [48,49]. Autophagy plays a critical role in maintaining cellular homeostasis by removing damaged organelles and proteins, thus preventing stress-induced apoptosis. Additionally, by recycling intracellular components, autophagy supplies nutrients and energy, enabling CD133-positive cells to survive in nutrient-deprived conditions. CD133-positive cells are also known to secrete interleukin-4, an immunomodulatory factor that further suppresses apoptosis [50]. These findings suggest that CD133-positive CSCs may serve as a prognostic marker of poor outcomes in CRC. In studies examining CD133 expression in primary CRC tissues, using immunohistochemistry to assess its association with OS, Kojima et al. analyzed 160 patients with stage I–IV CRC and reported that CD133 expression was associated with poor prognosis [51]. Similarly, Jao et al. conducted a study with 233 patients and reported that CD133 expression was associated with poorer prognosis [52]. Additionally, meta-analyses conducted by Chen and Wang reported similar findings [33,34]. Conversely, Kim et al. [11], in a study of 523 cases, found no significant association between CD133 expression in primary CRC tissues and OS. Regarding CD133-positive CTCs and prognosis, Pilati et al. analyzed 50 CRC cases with liver metastases and reported that CD133-positive cases had a poorer prognosis than did CD133-negative cases [53]. Similarly, Lin et al. investigated 100 patients with CRC and found that CD133-positive CTCs were a reliable predictor of recurrence and were associated with a shorter median OS [54]. However, Iinuma et al., in a study of 735 CRC cases, reported that CD133-positive CTCs alone were not significantly associated with OS [3].

The discrepancies in these findings may be due to differences in patient cohorts, study designs, CTC isolation methods, evaluation techniques, and cutoff values used in the analyses. Moreover, Zhou et al. reported that CTCs can be classified into epithelial and mesenchymal CTCs, and that CRC patients with elevated levels of CD133-positive mesenchymal CTCs were more prone to distant metastases and exhibited poorer PFS [55]. These findings suggest that the proportion of mesenchymal CTCs may also influence prognosis. Despite these variations, all reports indicate that CD133-positive CTCs are associated with prognosis to some extent. In our study, using the OncoQuick density gradient system to analyze 195 cases, we demonstrated a strong association between CD133-positive CTCs and prognosis. Furthermore, Ren et al. [27] reported that CD133-positive CTCs were a particularly reliable prognostic marker in middle-stage CRC. Consistent with this, our study also found a significant association between CD133-positive CTCs and prognosis in stage III cases (Appendix A).

CEA is recommended as a tumor marker for CRC in the NCCN guidelines [4,5,6,7,8,9]. Elevated CEA levels have been reported to be associated with poor prognosis [56]. Chu et al. demonstrated that the combination of CTC count and CEA levels serves as a reliable prognostic indicator. Specifically, cases with high CTC counts (≥4 CTCs per 2 mL of blood) and high CEA levels were associated with poor prognosis [57]. In our study, CEA levels were confirmed to be a reliable prognostic marker. However, there was no significant association between CEA levels and CD133 expression in CTCs, suggesting that they are independent prognostic factors. We further investigated the prognostic value of combining these independent predictors. Patients with both CD133-positive CTCs and high CEA levels had significantly poorer prognoses than did those without these factors. The HR for DSS was higher when both CD133-positive CTCs and high CEA levels were considered together than when either factor was considered alone, suggesting that their combination enhances prognostic accuracy. Several studies have reported that the combination of CD133-positive CTCs with other prognostic markers improves prognostic accuracy. Zahran et al. reported that patients with both CD133-positive and CD44-positive CTCs had significantly poorer OS [58]. Similarly, Iinuma et al. demonstrated that the combination of CEA, cytokeratin, and CD133 was a reliable prognostic marker for poor outcomes in Dukes’ stage B and C CRC cases [3]. Additionally, in patients with CRC and liver metastases, concurrent positivity for CD133, CD44, and CD54 was associated with poorer prognosis [59].

In our analysis of clinicopathological factors in primary CRC tumors, we found that CD133-positive CTCs were significantly associated with serosal involvement. Since CD133-positive cells are believed to originate exclusively from CD133-positive tumor cells, the presence of CD133-positive CTCs likely reflects CD133 expression in the primary tumor [27]. Numerous studies have reported that CD133 positivity in primary CRC is associated with tumor depth [20,33,60,61,62,63]. In vitro [64] and in vivo [65] studies have shown that CD133-positive CRC cells exhibit greater invasiveness than CD133-negative cells do. Chao et al. further demonstrated that CD133-positive CRC cells enhance their interaction with surrounding fibroblasts, which may contribute to their increased invasiveness relative to CD133-negative cells [29].

However, no significant correlation was found between distant metastasis and CD133-positive CTCs. Okada et al. highlighted differences in CD133 expression based on the metastatic site. They reported that CD133 expression levels in liver metastases were significantly higher than those in the corresponding primary tumors, whereas CD133 expression levels in peritoneal metastases were significantly lower than those in primary tumors [30]. Liver metastases are thought to occur because of the high EMT capacity of CD133-positive cells under hypoxic conditions. In contrast, peritoneal metastases are believed to result from the high expression of β1-integrin in CD133-negative cells [65]. Similarly, Gao et al. reported that different stem cell markers are involved in liver and lung metastases of CRC, indicating that chemoattraction and adhesion mechanisms differ according to the metastatic organ [66].

This study has several limitations. First, in stage II and III cases, the number of deaths was low, which limited the feasibility of multivariate analysis to evaluate the impact of CD133-positive CTCs on prognosis in these cases. However, in univariate analysis for stage III cases, CD133-positive CTCs appeared to be a potential risk factor for poor prognosis. Second, in this study, the assessment of CD133 expression in the blood relied solely on its presence or absence, without considering expression intensity. This may have contributed to discrepancies between our results and those of previous reports. Nonetheless, many studies have reported associations between CD133 expression and prognosis in various contexts, which aligns with our findings. Third, DSS analysis was conducted by censoring deaths unrelated to CRC; however, a competing risk model was not applied. This may introduce potential bias, and the application of such models is recommended in future studies.

## 4. Materials and Methods

### 4.1. Patients and Sample Collection

A total of 195 patients who underwent CRC resection at our institution from 2010 to 2020 were included in the study. Patients with synchronous or metachronous cancers were excluded from the analysis. In addition, blood samples from 10 healthy volunteers were obtained to serve as controls. For patients with colorectal cancer, blood was collected prior to surgical resection of the primary tumor to evaluate preoperative serum CEA levels and the presence of circulating tumor cells (CTCs), with a specific focus on CD133 expression. A CEA concentration of 5 ng/mL was used as the threshold for elevated levels. To reduce the potential contamination from skin epithelial cells, the initial 5 mL of peripheral blood was discarded. Thereafter, 20 mL of blood was collected and subjected to density gradient centrifugation using the OncoQuick system (Greiner Bio-One GmbH, Frickenhausen, Germany) in accordance with the manufacturer’s instructions. Circulating tumor cells were subsequently isolated and resuspended in 400 μL of phosphate-buffered saline.

As negative controls, peripheral blood samples obtained from healthy volunteers, confirmed to be free of epithelial cells, were processed using the same OncoQuick system. The study protocol was approved by the Research Ethics Committee of the University of Fukui (Approval No. 20200058), and written informed consent was obtained from all participants prior to enrollment and publication of the study results.

### 4.2. Reverse Transcription-Polymerase Chain Reaction

The total RNA was isolated from the enriched tumor cell fraction using a ISOGEN reagent (Nippon Gene, Tokyo, Japan), followed by reverse transcription into complementary DNA (cDNA) using the PrimeScript RT reagent kit (Takara Bio Inc., Otsu, Japan). The coding region of CD133 was amplified via polymerase chain reaction (PCR) using the following primer sequences: forward 5′-CAGAGTACAACGCCAAACCA-3′ and reverse 5′-AAATCACGATGAGGGTCAGC-3′, as previously described [67]. PCR amplification was carried out in a PTC-100 Programmable Thermal Controller (MJ Research Inc., Manahawkin, NJ, USA) under the following cycling conditions: 35 cycles of denaturation at 94 °C for 1 min, annealing at 55 °C for 1 min, and extension at 72 °C for 2 min. The resulting PCR products were purified using the QIAquick PCR Purification Kit (Qiagen, Hilden, Germany) and separated by electrophoresis on a 1.2% agarose gel. Bands corresponding to CD133 were visualized using ethidium bromide staining and subjected to sequencing to confirm identity. All PCR assays were performed in duplicate to ensure reproducibility, and semi-quantitative assessment of CD133 mRNA expression was based on band intensity. To avoid confusion in terminology, in this manuscript, “CD133-positive CTCs” are operationally defined as peripheral blood samples in which CD133 mRNA was detected using RT-PCR.

### 4.3. Clinical Assessment

Comprehensive clinical data, including patient demographics (age and sex), tumor characteristics (size, anatomical location, histological type, and depth of invasion), lymph node and distant metastasis status, TNM stage, preoperative serum CEA levels, and disease-specific survival (DSS), were collected for all patients diagnosed with colorectal cancer (CRC). DSS was defined as the interval from the date of surgical resection to CRC-related death, with deaths from other causes treated as censored events. Tumor histopathology and staging were assessed according to the TNM classification system. Postoperative surveillance included routine follow-up assessments consisting of tumor marker evaluations every 3 months, contrast-enhanced abdominal computed tomography every 6 months, and colonoscopy every 3 years.

### 4.4. Statistical Analysis

The relationship between CD133 expression in circulating tumor cells (CTCs) and clinicopathological factors was evaluated using the chi-square test for categorical variables and logistic regression for multivariate analysis. To assess the association between CD133 expression and disease-specific survival (DSS), as well as the prognostic impact of CEA levels and the combination of CD133 expression and CEA levels, DSS was analyzed using the Kaplan–Meier method, and group comparisons were conducted using the log-rank test. Hazard ratios (HRs) were calculated using the Cox proportional hazards regression model. All statistical analyses were performed using IBM SPSS Statistics version 21.0 (IBM Japan, Ltd., Tokyo, Japan). A *p*-value of <0.05 was considered statistically significant.

## 5. Conclusions

Our study confirmed that CD133 expression in CTCs is a reliable prognostic marker in CRC. Furthermore, combining CD133 expression in CTCs with the established tumor marker CEA effectively stratified patients into groups with favorable and unfavorable prognoses. This approach may aid treatment decisions, including the selection of candidates for postoperative adjuvant chemotherapy.

## Figures and Tables

**Figure 1 ijms-26-04740-f001:**
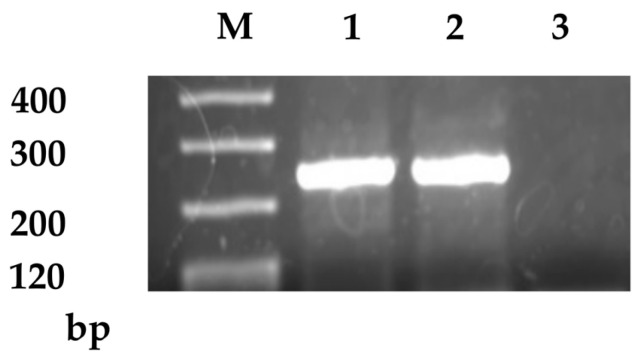
Representative image of CD133 mRNA expression detected by RT-PCR. Lanes 1 and 2 show positive expression of CD133 mRNA, while lane 3 shows negative expression. Lane M corresponds to the DNA size marker.

**Figure 2 ijms-26-04740-f002:**
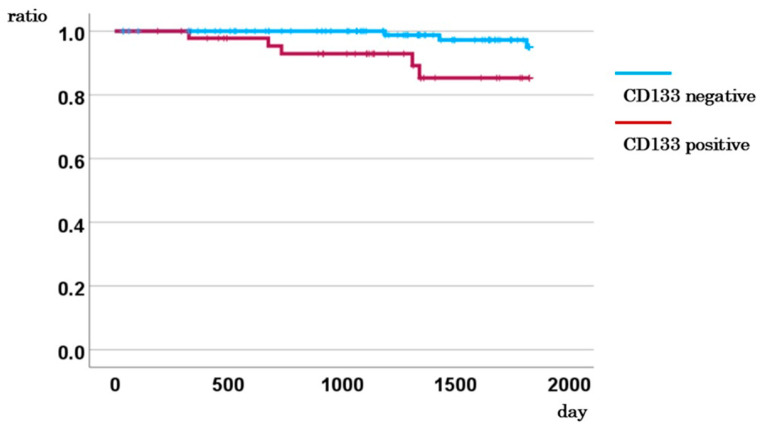
5-year DSS for CD133-positive and CD133-negative cases.

**Figure 3 ijms-26-04740-f003:**
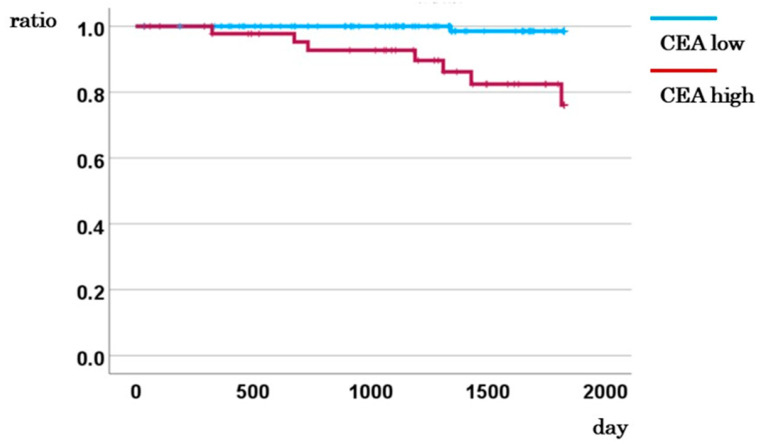
5-year DSS for cases with low and high CEA levels.

**Figure 4 ijms-26-04740-f004:**
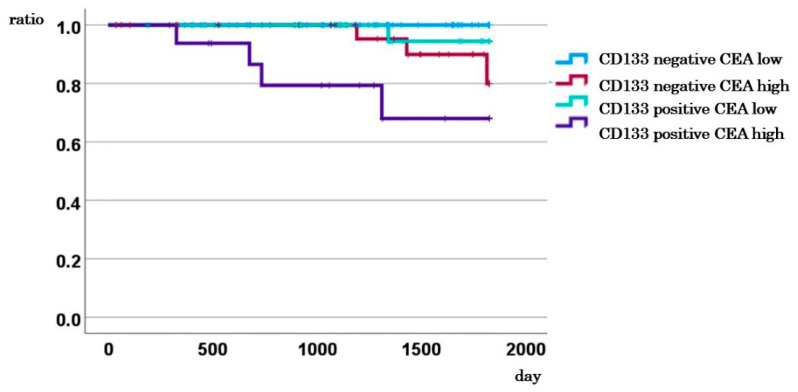
5-year DSS based on the combination of CD133 and CEA.

**Figure 5 ijms-26-04740-f005:**
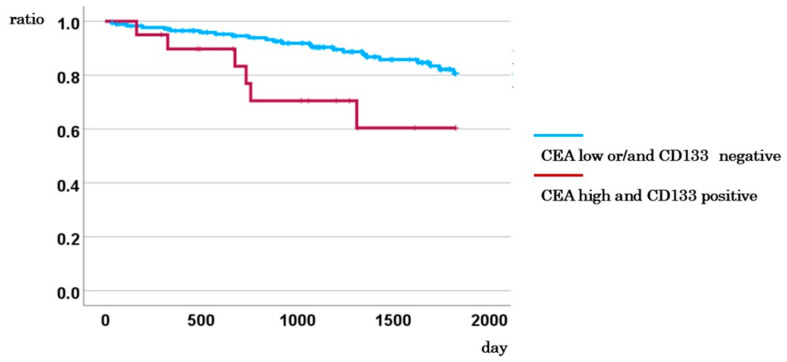
Comparison of the 5-year DSS between cases with CD133-positive CTCs and high CEA levels and other cases.

**Table 1 ijms-26-04740-t001:** Univariate analysis of the association between CD133 expression and clinicopathological variables.

			*CD133* mRNA		
		No. of Cases	Negative Cases (%)	Positive Cases (%)	*p*-Value
All cases (%)		195	142 (72.8)	53 (27.2)	
Age (median years)			71.0	70.0	0.900
Sex	Male	105	76 (72.4)	29 (27.6)	0.882
	Female	90	66 (73.3)	24 (26.7)	
Location	Right colon	74	53 (71.6)	21 (28.4)	0.769
	Left colon	121	89 (73.6)	32 (26.4)	
Size (average mm)	<50	113	88(77.9)	25(22.1)	0.062
	≧50	82	54(65.9)	28(34.1)	
Histological type	Differentiated	184	135 (73.4)	49 (26.6)	0.481
	Undifferentiated	11	7 (63.6)	4 (36.4)	
Serosal involvement	Negative	52	44 (84.6)	8 (15.4)	0.026
	Positive	143	98 (68.5)	45 (31.5)	
Lymph node metastasis	Negative	113	82 (72.6)	31 (27.4)	0.925
	Positive	82	60 (73.2)	22 (26.8)	
Distant metastasis	Negative	163	116 (71.2)	47 (28.8)	0.241
	Positive	32	26 (81.2)	6(18.8)	
Stage	I, II	108	78 (72.2)	30 (27.8)	0.834
	III, IV	87	64 (73.6)	23 (26.4)	
CEA	5.0>	125	92 (73.6)	33 (26.4)	0.744
	5.0≦	70	50 (71.4)	20 (28.6)	

**Table 2 ijms-26-04740-t002:** Multivariate analysis of the association between CD133 expression and clinicopathological variables.

		Multivariate		
	Variable	Odds Ration	95% CI	*p*-Value
Age		0.997	0.967–1.028	0.847
Sex	Male vs. Female	0.961	0.493–1.874	0.907
Serosal invasion	Negative vs. Positive	3.000	1.225–7.345	0.016
Lymph node metastasis	Negative vs. Positive	0.850	0.412–1.754	0.660
Distant metastasis	Negative vs. Positive	0.472	0.165–1.353	0.162
CEA	5.0> vs. 5.0≦	1.074	0.532–2.167	0.842

**Table 3 ijms-26-04740-t003:** Multivariate Cox analysis for DSS incorporating CD133 Expression.

			Multivariate	
	Variable	HR	95% CI	*p*-Value
Distant metastasis	Negative vs. Positive	35.713	14.520–87.838	<0.001
CD133 expression	Negative vs. Positive	3.057	1.298–7.196	0.011

**Table 4 ijms-26-04740-t004:** Multivariate Cox analysis for DSS incorporating CEA levels.

			Multivariate	
	Variable	HR	95% CI	*p*-Value
Distant metastasis	Negative vs. Positive	19.715	8.296–46.853	<0.001
CEA level	CEA < 5 vs. CEA ≧ 5	2.309	1.017–5.244	0.046

**Table 5 ijms-26-04740-t005:** Multivariate Cox analysis for DSS incorporating positive CD133 mRNA expression and high CEA levels.

			Multivariate	
	Variable	HR	95% CI	p-Value
**Distant metastasis**	Negative vs. Positive	33.112	13.797–79.468	**<0.001**
**CD133 and CEA**	Others vs. CD133 (positive) and CEA ≧ 5	5.948	2.210–16.007	**<0.001**

## Data Availability

All data included in this study are available upon request from the corresponding author.

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
