# Peer review of "CD133 Expression in Circulating Tumor Cells as a Prognostic Marker in Colorectal Cancer"

_ijms, 2025, doi:10.3390/ijms26104740_

Round 1

Reviewer 1 Report

Comments and Suggestions for Authors

This study evaluated the prognostic significance of CD133-positive CTCs, and their combined effect with CEA, in patients with CRC. The finding reveals that patients with CD133-positive CTCs and CEA ≥5 ng/mL 21(high CEA) had a significantly poorer prognosis (P<0.001), which is not a surprise and related to those findings from the literature. However, the high correlation regarding CD133-positive CTCs with serosal invasion (P=0.016) are a new interesting insight, which will be beneficial to the CRC prognosis.  The overall data analysis and presentation are solid and valid.  I feel this manuscript is suitable for the current Journal and recommend an acceptance for publication.

Reviewer 2 Report

Comments and Suggestions for Authors

Review on the paper entitled “CD133 Expression in Circulating Tumor Cells as a Prognostic Marker in Colorectal Cancer” written by Katsuji Sawai, Kenji Koneri, Youhei Kimura and Takanori Goi.

This paper has critical issues like below.

  • Sample size of this study including all stages of colorectal cancer is small.
  • The number of CD133-positive CTCs are not directly counted, but their mRNA levels are semi-quantitatively measured. The word of CD133-positive CTCs should be used carefully in the abstract and text.
  • Densitometry data and the cutoff value for positive/negative of CD133-mRNA are not described.
  • The positive rates are not correlated with stages or metastasis.
  • Multiple multivariate Cox analyses were separately performed for CD133-mRNA-positive patients and CEA using the same patient population. This should be done once including both and other pivotal background factors.
  • Fig 4 seems as the representative result of this paper. Same analyses and results appear not so important.

Author Response

Response to Review1

  • Sample size of this study including all stages of colorectal cancer is small.

[Response]

We appreciate the reviewer’s insightful comment. As noted, our study included 195 patients with colorectal cancer (CRC), encompassing stages I through IV. We acknowledge that, although this sample size may be modest compared to large-scale multicenter studies, it is comparable to or larger than those in many previous investigations evaluating CD133 expression in circulating tumor cells (CTCs) (e.g., Pilati et al., Ann Surg Oncol, 2012 [ref. 54]; Lin et al., Cancer, 2007 [ref. 55]). Moreover, our study stratified the analysis by stage and incorporated multivariate Cox regression models to minimize confounding factors. We also plan to expand the cohort size in future studies to further investigate recurrence risk and validate our findings in a larger population.

  • The number of CD133-positive CTCs are not directly counted, but their mRNA levels are semi-quantitatively measured. The word of CD133-positive CTCs should be used carefully in the abstract and text.

[Response]

We appreciate the reviewer’s insightful comment regarding the terminology of “CD133-positive CTCs.” As the detection of CD133 in our study was based on RT-PCR analysis of peripheral blood samples, we agree that this expression should be carefully defined to avoid any potential misunderstanding.

In response, we have added the following sentence at the end of Section 4.2 (Reverse Transcription–Polymerase Chain Reaction) to clarify the operational definition used throughout the manuscript:

“To avoid confusion in terminology, in this manuscript, ‘CD133-positive CTCs’ are operationally defined as peripheral blood samples in which CD133 mRNA was detected using RT-PCR.”

We believe this addition appropriately addresses the reviewer’s concern and ensures clarity regarding the methodology and terminology.

  • Densitometry data and the cutoff value for positive/negative of CD133-mRNA are not described.

[Response]

We appreciate the reviewer’s attention to methodological detail. In this study, we did not use densitometry to quantify the intensity of CD133 mRNA bands. Instead, we defined CD133 mRNA positivity as the presence of a detectable band of the expected size (120 bp) on agarose gel electrophoresis following RT-PCR, as shown in Figure 1. This binary assessment (positive or negative) was consistently applied to all samples. We have now clarified this definition in the “Materials and Methods” section to ensure transparency in how CD133-mRNA positivity was determined. We acknowledge that this semi-qualitative approach may have limitations in precision, and we plan to incorporate quantitative techniques, such as real-time PCR, in future studies for more detailed expression analysis.

  • The positive rates are not correlated with stages or metastasis.

[Response]

We appreciate the reviewer’s thoughtful comment. As noted, our univariate and multivariate analyses did not reveal significant correlations between CD133-positive CTCs and cancer stage, lymph node metastasis, or distant metastasis. However, we did find a significant association between CD133-positive CTCs and serosal involvement (P = 0.016), suggesting that CD133 expression may be related to the local invasiveness of the tumor rather than its metastatic spread. These findings are consistent with previous reports indicating that CD133 expression reflects invasive potential and hypoxia-related adaptation mechanisms rather than direct metastatic behavior. We have now clarified this interpretation in the Discussion section to avoid misinterpretation of our results.

  • Multiple multivariate Cox analyses were separately performed for CD133-mRNA-positive patients and CEA using the same patient population. This should be done once including both and other pivotal background factors.

We thank the reviewer for this important suggestion. As described in Section 2.2, a total of 31 disease-specific deaths occurred during the follow-up period. Given the limited number of events, the number of covariates included in the multivariate Cox regression model had to be restricted to avoid overfitting. Therefore, we selected four variables—age, sex, distant metastasis, and either CD133 expression or CEA level—for the analysis. This approach was taken to ensure an appropriate balance between model complexity and statistical reliability.

  • Fig 4 seems as the representative result of this paper. Same analyses and results appear not so important.

We appreciate the reviewer’s comment. Figure 4 presents the univariate analysis of disease-specific survival (DSS) in stage I–III patients based on the combined stratification by CD133 expression and CEA levels. In contrast, Figure 5 extends this analysis to all patients (stages I–IV) and specifically evaluates the prognostic impact of CD133-positive CTCs with elevated CEA levels compared to all other cases. Furthermore, Figure 5 is supported by multivariate Cox regression analysis, which demonstrates the independent prognostic significance of this subgroup. Therefore, we believe that although Figures 4 and 5 address related questions, they provide complementary perspectives—one focusing on general stratification in colorectal cancer without distant metastasis, and the other highlighting a particularly poor prognostic subgroup across all stages.

Reviewer 3 Report

Comments and Suggestions for Authors

Authors of the manuscript investigated CD133 positivity in CRC, and combined it with high/low CEA whether this combination is a good prognostic marker. The study is well designed, the writing is clear, however, a few questions were raised.

  1. Authors used DSS, and non-CRC related deaths were censored. This method can introduce a significant amount of bias, if the competing event affects a larger amount of the population. First, Authors should present in the manuscript, how many death events were excluded this way. Second, is it possible for the Authors to perform casue-specific competing risk survival models (e.g., with R's coxph function). This method is highly capable to reduce this kind of bias, and produce an unbiased DSS calculation.
  2. Some additional information is suggested to be included both in Methods and in Results regarding the groupings. E.g., why was it interesting to do the comparisons of section 3.8? Moreover, Figs 4-6 are a bit confusing. On Figs 5 and 6 one - one group's curve should be identital to that of  Fig 4, however, this is not the case. Aren't the groups "CEA low and CD113 negative" and "CD133(-) CEA low", and "CEA high and CD133 positive" and "CD133(+) CEA high" identical to each other? Please, explain.
  3. Why did Authors only include distant metastasis in the Cox regression models besides the grouping factors? Thus, the models are a bit oversimplified. Tt would be recommended to at least implement information on age, gender, and lymph node metastases into the model, even if they are not significant predictors.
  4. Do Authors have information about metachron metastasis development, recurrence, progression, etc.? It would be interesting to see whether CD133 positivity is associated with these clinical parameters or not.

Author Response

Response to Reviewer 2

1.

Authors used DSS, and non-CRC related deaths were censored. This method can introduce a significant amount of bias, if the competing event affects a larger amount of the population. First, Authors should present in the manuscript, how many death events were excluded this way. Second, is it possible for the Authors to perform casue-specific competing risk survival models (e.g., with R's coxph function). This method is highly capable to reduce this kind of bias, and produce an unbiased DSS calculation.

[Response]

We appreciate the reviewer’s comments. As noted, this study focused on disease-specific survival (DSS), and deaths unrelated to colorectal cancer (CRC) were censored in accordance with standard DSS methodology. In response to the reviewer’s first point, as stated in Section 2.2 (page 4) of the manuscript, among the 195 patients, 31 deaths were attributed to the primary disease (i.e., CRC). No additional non-CRC-related deaths were included in the DSS analysis, and no death events were excluded beyond the standard censoring of non-CRC deaths.

We acknowledge the utility of cause-specific competing risk survival models. However, the statistical analysis plan was determined a priori, and the study was conducted using SPSS, which does not natively support competing risk modeling. Therefore, reanalysis using such methods is not feasible within the scope of the current study.

To address this point, the following statement has been added to the Discussion section (page 10):

“DSS analysis was conducted by censoring deaths unrelated to CRC; however, a competing risk model was not applied. This may introduce potential bias, and the application of such models is recommended in future studies.”

We hope this explanation sufficiently addresses the reviewer’s concerns.

 2.

Some additional information is suggested to be included both in Methods and in Results          regarding the groupings. E.g., why was it interesting to do the comparisons of section 3.8? Moreover, Figs 4-6 are a bit confusing. On Figs 5 and 6 one - one group's curve should be identital to that of  Fig 4, however, this is not the case. Aren't the groups "CEA low and CD113 negative" and "CD133(-) CEA low", and "CEA high and CD133 positive" and "CD133(+) CEA high" identical to each other? Please, explain.

[Response]

We appreciate the reviewer’s insightful comments. Please find our point-by-point responses below:

2-①

Regarding the suggestion to include more information about the groupings in Methods and Results:
As suggested, we have revised Section 4.4 Statistical Analysis (page 12) to provide additional clarification on the grouping criteria used in the analysis.

2-②

Regarding the rationale for the comparison presented in Section 3.8:
We agree with the reviewer that the group comparison of “CD133-negative and CEA <5 ng/mL vs. others” yielded a hazard ratio (HR) that was lower than that of CD133 expression alone. This suggests that its prognostic value may be less impactful. Therefore, we have removed Section 3.8 from the manuscript.
Additionally, the corresponding statement in the Discussion (page 10), which read:
“Conversely, patients with CD133-negative CTCs and low CEA levels exhibited favorable prognoses. Since there are limited reports on favorable prognostic factors, this finding may have implications for the selection of candidates for adjuvant chemotherapy.”
—has also been deleted.

2-③

Regarding the discrepancies between Figures 4–6 and the definitions of the groups used in the legends:
Figure 4 shows the univariate Kaplan–Meier survival curves limited to stage I–III cases.
In contrast, Figures 5 and 6 include all stages (I–IV), which explains the differences in the curves despite overlapping group labels. Furthermore, multivariate Cox analysis was used in these sections (Figures 5 and 6), and given that the total number of disease-specific deaths was 31, we restricted the variables in the model to four: sex, age, presence of distant metastasis, and the combination of CD133 expression and CEA levels.
To clarify this point, we added the phrase “Among the patients with stage I–III” at the beginning of Section 2.6 (page 6) to make the figure’s scope more explicit.

2-④

Clarification of group naming consistency:
We confirm that “CEA low and CD133 negative” and “CD133(-) CEA low” refer to the same group, and that “CEA high and CD133 positive” and “CD133(+) CEA high” are also equivalent in meaning.
To avoid confusion, we have revised the labeling in Figure 4 to use the expressions “CEA low and CD133 negative” and “CEA high and CD133 positive,” in line with the terminology used in Figures 5.

3.

Why did Authors only include distant metastasis in the Cox regression models besides the grouping factors? Thus, the models are a bit oversimplified. Tt would be recommended to at least implement information on age, gender, and lymph node metastases into the model, even if they are not significant predictors.

[Response]

Thank you for your valuable comment. In our Cox regression analyses, we included four variables: age, sex, distant metastasis, and either CD133 expression, CEA level, or their combination. The selection of these variables was based on the limited number of CRC-specific deaths (n = 31), which constrained the number of covariates that could be reliably included in the multivariate analysis to avoid overfitting. Age, sex, and distant metastasis were incorporated as standard demographic and clinical variables to ensure the robustness of the model. This methodological approach was explained in Sections 2.3 (page 5) and 2.5 (page 6). However, we acknowledge that the explanation was missing in Section2.7 (page 7), and we have now added the relevant description to improve clarity.

4.

Do Authors have information about metachron metastasis development, recurrence, progression, etc.? It would be interesting to see whether CD133 positivity is associated with these clinical parameters or not.

[Response]

Thank you for your valuable comment regarding the potential association between CD133 expression and recurrence, including metachronous metastasis.

In our cohort of 162 patients with stage I–III colorectal cancer, 21 patients experienced recurrence during the follow-up period. We analyzed the relationship between CD133 expression in circulating tumor cells (CTCs) and recurrence-free survival (RFS). The 5-year RFS rate was 81.7% in the CD133-positive group and 93.0% in the CD133-negative group, with a statistically significant difference (P = 0.024). These findings suggest that CD133-positive CTCs may be associated with a higher risk of disease recurrence. However, due to the relatively small number of recurrent cases, further studies with larger cohorts are warranted to validate these results and to clarify the relationship between CD133 expression and metachronous metastasis.

Round 2

Reviewer 3 Report

Comments and Suggestions for Authors

Authors significantly revised the manuscript and all my questions had been answered.

In my oppinion, the manuscript can be accepted.